# Harnessing Venetoclax in NPM1-Mutated AML: A Path to Sustained Remission and Beyond

**DOI:** 10.3390/cancers17233733

**Published:** 2025-11-21

**Authors:** Matteo Molica, Claudia Simio, Laura De Fazio, Caterina Alati, Massimo Martino, Marco Rossi

**Affiliations:** 1Division of Hematology-Oncology, Azienda Universitaria Ospedaliera Renato Dulbecco, 88100 Catanzaro, Italy; 2Hematology and Stem Cell Transplantation and Cellular Rerapies Unit (CTMO), Department of Hemato-Oncology and Radiotherapy, Grande Ospedale Metropolitano “Bianchi-Melacrino-Morelli”, Presidio Morelli, 89128 Reggio Calabria, Italy

**Keywords:** NPM1-mutated AML, Venetoclax, allo-HSCT, measurable residual disease (MRD), treatment-free remission (TFR)

## Abstract

Acute myeloid leukemia (AML) with nucleophosmin 1 (NPM1) mutation represents a distinctive subpopulation with unique molecular and clinical features. The introduction of venetoclax in combination with hypomethylating agents has revolutionized the treatment of this setting, especially in patients unfit for intensive chemotherapy. This review analyses the most recent evidence on the efficacy and safety of venetoclax in patients with NPM1-mutated AML, exploring the biological mechanisms underlying sensitivity, post-remission strategies, and therapeutic discontinuation options. Key unresolved clinical questions and future perspectives are also discussed, focusing on ongoing trials and the potential for personalized management based on minimal residual disease monitoring. The aim is to provide an updated and critical overview to support clinicians in therapeutic decisions and to promote further targeted research.

## 1. Introduction

Acute myeloid leukemia (AML) harboring nucleophosmin 1 (NPM1) mutations represents a biologically and clinically distinct subset, occurring in approximately one-third of adult patients and in over half of those with a normal karyotype [1]. In this review, we primarily focus on isolated NPM1-mutated AML, where the prognostic impact of the mutation and the sensitivity to venetoclax-based regimens are most clearly defined. However, it is recognized that co-occurring mutations are frequent, particularly FLT3-ITD, TP53, ASXL1, and RUNX1, and these may modify both response to therapy and relapse risk [2]. Therefore, when study cohorts include mixed molecular profiles, outcomes for NPM1-mutated patients with high-risk co-mutations are discussed separately, highlighting where therapeutic expectations diverge from those of isolated NPM1-mutated disease.

Nevertheless, a significant proportion of NPM1-mutated patients eventually relapse, often heralded by the reappearance of measurable residual disease (MRD), which carries adverse prognostic implications [3].

In recent years, the introduction of venetoclax, a selective BCL-2 inhibitor, has revolutionized the therapeutic landscape of AML, particularly for patients deemed unfit for intensive chemotherapy. In combination with hypomethylating agents (HMAs) or low-dose cytarabine, venetoclax has demonstrated remarkable efficacy in inducing rapid and deep remissions [4]. This benefit appears especially pronounced in patients with NPM1 mutations, who exhibit heightened sensitivity to venetoclax-based therapy and achieve high rates of early molecular responses [5].

Emerging evidence from both clinical trials and real-world studies highlights the prognostic significance of achieving MRD negativity within the early treatment cycles—most notably by the fourth cycle, when the majority of deep responses occur [6]. MRD assessment is therefore gaining recognition not only as a key survival biomarker but also as a potential decision-making tool for tailoring therapy duration. Throughout the manuscript, we refer to MRD as molecular MRD assessed by quantitative RT-PCR for NPM1. Flow cytometry MRD is mentioned only when specifically noted.

Concurrently, a novel therapeutic paradigm is taking shape: the possibility of safely discontinuing therapy in carefully selected patients who achieve deep and sustained molecular remission. Within this framework, the concept of treatment-free remission (TFR) is emerging as a realistic therapeutic goal, at least for a subset of NPM1-mutated AML patients treated with venetoclax-based regimens [7].

In light of these developments, this review aims to critically analyze the clinical and molecular efficacy of venetoclax in NPM1-mutated AML, with a particular focus on MRD-guided discontinuation strategies and their potential to redefine long-term disease management.

## 2. Molecular Rationale for Venetoclax Sensitivity in NPM1-Mutated AML

To fully elucidate the efficacy of venetoclax in patients with NPM1-mutated AML, it is crucial to consider the distinctive molecular context defining this subgroup. The NPM1 mutation leads to a characteristic gene expression profile marked by aberrant overexpression of HOX genes—particularly the *HOXA* and *HOXB* clusters—which play a pivotal role in regulating hematopoietic cell differentiation and survival [8]. This transcriptional signature drives leukemic cells to rely heavily on the anti-apoptotic protein BCL-2, thereby conferring heightened sensitivity to selective BCL-2 inhibition by venetoclax [9] (Figure 1).

However, anti-apoptotic dependency among leukemic cells is not uniform. In certain sub-clones, cellular survival is predominantly mediated by MCL-1 rather than BCL-2. Such heterogeneity may influence the depth and durability of therapeutic responses, as compensatory upregulation of MCL-1 represents a well-established mechanism of secondary resistance to venetoclax [10]. This dynamic molecular interplay suggests that future treatment strategies could incorporate dual inhibition of BCL-2 and MCL-1 to enhance molecular clearance and mitigate relapse risk.

Recent omics-based studies have further refined our understanding of apoptotic regulation in NPM1-mutated AML. Single-cell RNA sequencing (scRNA-seq) and proteomic analyses have demonstrated heterogeneous expression patterns of BCL-2 family members across leukemic sub-clones, confirming that NPM1-mutated blasts display predominant BCL-2 dependency compared with MCL-1 or BCL-XL pathways [9]. These findings reinforce the biological rationale for selective BCL-2 inhibition and pave the way for personalized therapeutic combinations tailored to each patient’s apoptotic dependency profile.

## 3. Clinical and Molecular Efficacy of Venetoclax in NPM1 Mutated AML

The evidence summarized in this section refers predominantly to patients with isolated NPM1 mutations, as these represent the population with the most consistent and favorable responses to venetoclax-based therapy. When available, outcomes for NPM1-mutated patients with co-mutations, particularly FLT3-ITD (high allelic ratio), TP53, ASXL1, or RUNX1, are presented separately, given their distinct molecular biology and generally lower rates of durable molecular remission. [11].

The NPM1 mutation, which can now be easily monitored through molecular assays, has emerged not only as a prognostic biomarker but also as a potential predictor of response to venetoclax-based therapy. Multiple prospective and retrospective studies have consistently demonstrated the marked sensitivity of NPM1-mutated leukemic cells to BCL-2 inhibition, leading to high rates of deep and durable molecular remissions—sometimes enabling safe treatment discontinuation [9].

It is also important to consider the clonal context of the NPM1 mutation. In most patients, NPM1 is a founding clonal event associated with high BCL-2 dependency and strong venetoclax sensitivity; however, when NPM1 occurs as a subclonal lesion—often alongside high-risk co-mutations—apoptotic dependency may shift toward MCL-1 or BCL-XL, potentially reducing the depth and durability of response.

This section reviews the principal clinical settings in which venetoclax-based regimens are employed in NPM1-mutated AML, focusing on overall response rates, MRD clearance, remission duration, and the emerging possibility of planned therapy discontinuation.

### 3.1. Venetoclax-Based Regimens in the Frontline and MRD-Directed Settings

One of the clinical contexts in which the combination of venetoclax and HMAs has shown the most consistent benefit is the frontline treatment in patients who are ineligible for intensive chemotherapy. In this setting, the venetoclax–azacitidine (VEN–AZA) regimen has yielded remarkably high response rates, particularly among patients harboring NPM1 mutations [12,13]. Retrospective analyses from the MD Anderson Cancer Center reported complete remission (CR) or composite CR (CR/CR with incomplete count recovery) rates exceeding 95%, with a 12-month overall survival (OS) approaching 80% and a favorable toxicity profile compared with intensive chemotherapy—supporting the strong efficacy and tolerability of this combination, even in older or frail patients [14].

A pivotal recent contribution comes from Othman et al. [12], who analyzed 76 patients with NPM1-mutated AML treated with venetoclax combined with either HMAs or LDAC. In this cohort, 58% achieved bone-marrow molecular MRD negativity and an additional 18% had a ≥4-log reduction in NPM1 transcripts as best response. MRD negativity by the end of cycle 4 was the strongest predictor of survival (2-year OS 84% in MRD-negative vs. 46% in MRD-positive patients). Moreover, 22 patients who electively discontinued venetoclax-based therapy while in sustained MRD-negative remission (median 8 cycles) achieved a 2-year treatment-free remission of 88%.

Similarly, a prospective study conducted by Di Nardo et al. demonstrated that the VEN–AZA combination can induce deep molecular responses in patients with IDH2- or NPM1-mutated AML [15]. Importantly, elective treatment discontinuation after at least 12 months of sustained MRD negativity proved feasible, with limited and manageable relapse rates.

Beyond the frontline setting, venetoclax-based regimens have gained increasing relevance in patients with molecular relapse or persistent MRD after prior intensive chemotherapy. In a multicenter study, venetoclax—administered in combination with either an HMA or LDAC—achieved MRD negativity in 71% of patients with “molecular failure,” often within two to three cycles. In this context, venetoclax not only prevented overt hematologic relapse but also served as an effective bridge to allogeneic stem cell transplantation (HSCT), improving the molecular status before HSCT [16].

Finally, comparative studies have evaluated VEN–HMA regimens versus intensive chemotherapy in patients aged ≥60 years with NPM1-mutated AML, showing that low-intensity strategies are non-inferior—and potentially superior—in selected subgroups [17]. Collectively, these findings are prompting a re-evaluation of the role of intensive chemotherapy even in relatively younger NPM1-mutated patients, reinforcing the paradigm shift toward less toxic, molecularly guided therapeutic approaches.

### 3.2. Overall Response, MRD Clearance and Duration of Remission

From an efficacy standpoint, patients with NPM1-mutated AML represent a subgroup with exceptionally high response rates to venetoclax-based therapy. Across multiple clinical studies, CR or composite remission has been achieved in over 70–80% of cases, often accompanied by rapid and profound molecular responses [18].

In the study by Othman et al., 25% of patients achieved MRD negativity after only two treatment cycles, 47% after four cycles, and approximately 50% by the sixth cycle [12]. Comparable findings were reported by Jimenez-Chillon et al., who observed similarly rapid MRD clearance even in patients treated for molecular relapse [16]. These results confirm that deep molecular remission can be achieved early, even with non-intensive regimens, supporting MRD clearance as a potential surrogate endpoint for long-term outcomes.

Another key observation emerging from these studies concerns the possibility of safely discontinuing therapy in patients who achieve sustained MRD negativity. In several reports, patients who elected to discontinue treatment after achieving a complete molecular response maintained durable remission [19]. Similarly, in patients with molecular relapse, discontinuation after approximately 10 months of therapy was associated with a 62% probability of maintaining remission at two years [12].

It is important to note that TFR has primarily been observed in patients with isolated NPM1 mutation. In contrast, patients harboring high-risk co-mutations show lower rates of sustained molecular remission and higher relapse rates after discontinuation and therefore are not currently considered optimal candidates for venetoclax withdrawal strategies.

### 3.3. Toxicity and Adverse Events Associated with Venetoclax-Based Regimens

While venetoclax in combination with hypomethylating agents has substantially improved clinical outcomes in AML, this regimen is associated with a distinct toxicity profile that requires careful management. The most frequent adverse event is prolonged cytopenia, particularly grade ≥3 neutropenia [4], which may persist beyond induction and often necessitates dose modifications or temporary treatment interruptions. Such neutropenia contributes to an elevated risk of infectious complications, including bacterial and invasive fungal infections, warranting antimicrobial prophylaxis and close clinical monitoring. In routine practice, shortening venetoclax exposure per cycle (e.g., to 14–21 days) after achieving remission is commonly adopted to mitigate these risks [16].

Additional adverse effects include thrombocytopenia, anemia, and fatigue, primarily reflecting marrow suppression. Tumor lysis syndrome is uncommon but may occur in patients with high disease burden, underscoring the importance of early risk stratification and preventive measures. Overall, the toxicity profile of VEN + HMA regimens is manageable but demands individualized dosing strategies and dynamic hematologic monitoring. This is particularly relevant in older or frail patients, in whom maintaining adequate blood count recovery is essential for treatment continuity and preserving quality of life [15].

## 4. Post-Remission Strategies in NPM1-Mutated Patients: Observation, Maintenance, Transplant

The remarkable efficacy of the combination regimen has brought renewed attention to critical aspects of post-remission management, particularly regarding the optimal duration of therapy and the potential for treatment discontinuation in selected patients. Equally important is the need to better delineate those clinical scenarios in which intensive consolidative approaches, such as HSCT, remain indicated (Table 1).

These questions have become increasingly relevant as a growing proportion of patients now achieve complete molecular remission, often with limited toxicity and without the need for intensive chemotherapy. Addressing them requires a refined post-remission risk stratification that integrates clinical and molecular parameters, aiming to balance the potential benefits of consolidation (either transplant or maintenance) against the risks of toxicity, relapse, or overtreatment.

In this regard, the 2024 European Leukemia Net (ELN) recommendations reaffirm that the presence of an NPM1 mutation, in the absence of high-allelic ratio of FLT3-ITD, defines a favorable-risk subgroup. They also emphasize the pivotal role of molecular MRD assessment to inform post-remission therapeutic decisions [23]. Similarly, the 2024 National Comprehensive Cancer Network (NCCN) AML guidelines advocate for the systematic use of sensitive techniques such as quantitative real-time polymerase chain reaction (RT-PCR) for NPM1 to evaluate MRD and support a personalized, risk-adapted approach based on both molecular and clinical profiles [20].

### 4.1. Active Observation in MRD-Negative Patients

Among post-remission strategies, active observation of patients in complete molecular response is emerging as a more and more considered option, especially in the absence of additional genetic risk factors. Several studies [7,12,14] showed that a planned discontinuation of venetoclax-based therapy in patients with MRD negativity maintained for at least 12 months is feasible and safe, with only a minority of molecular relapses manageable by re-challenge. These data align with observations from multicenter international registries, such as those of the MD Anderson Cancer Center and studies aggregated by the Beat AML project, which emphasize the importance of MRD monitoring to identify patients who can be followed with a “watch and wait” strategy and avoid unnecessary further treatments [24]

### 4.2. The Controversial Role of Maintenance with Venetoclax

Maintenance with venetoclax, especially at low doses or with intermittent schedules, remains an area not yet defined in NPM1-mutated patients in molecular remission. Some retrospective data, including those derived from SEER and CIBMTR registries, suggest that this strategy could prolong response duration without significantly increasing hematologic toxicity, but randomized studies confirming its clinical benefit are still lacking. Ongoing trials like VIALE M are evaluating the role of maintenance with oral azacitidine and venetoclax in secondary AML or high-risk myelodysplastic syndromes, but its specific applicability in NPM1-mutated patients remains to be clarified [25]. At present, the 2024 NCCN guidelines emphasize that, in the absence of persistent MRD, maintenance should be considered an empirical rather than evidence-based option [20]. In parallel, oral formulations of HMAs have gained attention as feasible maintenance options. Oral azacitidine, evaluated in the QUAZAR AML-001 trial [26], has demonstrated improved OS in post-remission settings and is now being explored in combination with venetoclax to extend remission duration in NPM1-mutated AML. Such regimens may offer a convenient, outpatient-based approach to maintenance therapy, particularly for patients achieving molecular remission without the need for immediate transplant.

In clinical practice, several parameters may support the selection of patients who could benefit from maintenance therapy rather than simple observation. These include:(i)incomplete or fluctuating MRD response,(ii)delayed hematologic recovery requiring cautious dose adjustments,(iii)co-mutations associated with increased relapse risk (e.g., FLT3-ITD high allelic ratio, TP53, ASXL1, RUNX1), and(iv)prior history of early molecular relapse.

Conversely, patients with isolated NPM1 mutation, sustained MRD negativity for ≥6–12 months, and stable hematologic recovery may be considered appropriate candidates for observation and potential treatment discontinuation.

### 4.3. Allogeneic Transplant: Selective Indications

Allogeneic transplant continues to represent a crucial therapeutic resource in AML due to the graft-versus leukemia effect, but its indication must be weighed in light of associated toxicity and the molecular response achieved. In NPM1-mutated patients who reach MRD negativity, multicenter studies, including those by the NILG group and studies published by Fraccaroli et al., suggest that the additional benefit of transplant may be marginal [21,22]. In a cohort of 89 NILG patients, the 3-year OS for MRD negative transplanted patients was similar to that of non-transplanted patients (89% vs. 81%), with a nonsignificant difference in disease free survival (DFS) (80% vs. 75%) [22]. Similarly, an analysis of 174 transplanted NPM1-mutated patients by Fraccaroli et al. showed that pre-transplant MRD positivity increases relapse risk, but does not significantly affect OS and leukemia free survival (LFS), indicating that transplant may at least partially offset the unfavorable prognosis given by persistent MRD [21].

These data are supported by international experiences and European Society for Blood and Marrow Transplantation (EBMT) [22] guidelines, which recommend carefully considering MRD status in choosing the timing and intensity of conditioning in allogeneic transplant, aiming to optimize outcomes and reduce toxicity.

In conclusion, management of the NPM1-mutated patient in post-remission is evolving toward a personalized model, based on integrating molecular profile, MRD status, and clinical features. The increasing use of highly effective therapies such as venetoclax allows reconsideration of the need for allogeneic transplant, reserving it for patients with persistent MRD or associated high-risk mutations (e.g., high-allelic ratio of FLT3-ITD, TP53, RUNX1), in accord with the latest evidence from American and international multicenter studies [11,12].

## 5. Towards Treatment Discontinuation: Emerging Clinical Criteria for Venetoclax Withdrawal in NPM1-Mutated AML

The concept of TFR is becoming a clinical goal no longer reserved only for chronic myeloid leukemia, but now potentially achievable even in selected subsets of AML, including patients with NPM1 mutation treated with venetoclax-containing regimens. The evidence supporting this possibility comes from a series of observational, retrospective, and real-world studies which, despite the methodological limitations typical of these designs, suggest emerging criteria for identifying patients suitable for therapeutic discontinuation [27].

The clinical parameter most consistently associated with the possibility of discontinuation is the depth of molecular response, measured by quantitative PCR on bone marrow (Figure 2). In patients with NPM1 mutation, achieving MRD negativity has been correlated with a significant reduction in relapse risk and a prolonged remission duration, especially if molecular negativity is sustained over time. In addition to the depth of response, the duration of treatment prior to discontinuation also appears crucial. In most published experiences, patients who benefited from therapy discontinuation had received at least eight cycles of treatment with venetoclax and HMAs, suggesting that molecular negativity alone is not sufficient unless it is sustained over time [28]. Another crucial factor in selecting patients for discontinuation is the associated genetic profile. It is now evident that the presence of high-risk co-mutations, such as FLT3-ITD with high allelic burden, TP53, ASXL1 or RUNX1, can compromise the depth and duration of molecular response, even in the presence of NPM1 mutation [29]. In the available data, patients carrying these co-mutations show a lower rate of MRD negativity and a higher incidence of early relapse after discontinuation [12]. Therefore, their presence is generally considered a relative contraindication to therapeutic withdrawal, at least until specific data emerge in this regard.

Another important aspect is the management of post-discontinuation follow-up, which must necessarily include frequent and systematic molecular monitoring, ideally monthly during the first six months and every two months thereafter. The rise in NPM1 transcript levels can precede clinical relapse by several weeks, allowing timely intervention in the event of molecular recurrence. In reported cases, patients who relapsed molecularly after discontinuation and were re-challenged with venetoclax showed high response rates again, suggesting that therapy interruption does not necessarily compromise the drug’s effectiveness in the event of disease recurrence [19,30,31].

Ultimately, discontinuation of venetoclax treatment in NPM1-mutated AML patients is now emerging as a concrete and safe option in a selected subgroup of patients, defined by biological (stable MRD negativity, absence of high-risk mutations), clinical (good tolerance and hematologic recovery), and temporal (adequate therapy duration) criteria. 

Venetoclax-based treatment may be so highly effective that it can also be planned time-limited, thereby modifying the therapeutic paradigm that any treatment interruption is associated with a high risk of relapse.

Current evidence is predominantly derived from retrospective cohorts and real-world studies, and several prospective trials are underway to better define the criteria and safety of venetoclax discontinuation in NPM1-mutated AML. In particular, the multicenter phase II study NCT04867928 [32] is evaluating MRD-guided venetoclax + azacitidine in molecular relapse, while the GIMEMA AML2521 [33] trial is assessing venetoclax-based strategies to prevent hematologic relapse in patients with persistent MRD. Moreover, the phase III trial NCT06852222 [34] combining venetoclax, azacitidine and the menin inhibitor bleximenib aims to deepen molecular remission and potentially expand eligibility for treatment discontinuation. The results of these trials will be critical to establish standardized, evidence-based TFR criteria.

Therefore, we emphasize that TFR should currently be considered only in carefully selected patients, in whom MRD negativity is stable over time, hematologic recovery is adequate, and high-risk co-mutations are absent. We agree that prospective, randomized studies are needed to validate standardized TFR criteria, and the results of ongoing MRD-directed trials will be crucial in establishing evidence-based clinical algorithms.

## 6. Future Directions in the Management of NPM1-Mutated AML with Venetoclax-Based Regimens

Several ongoing clinical trials are exploring strategies to enhance the depth and durability of molecular response in NPM1-mutated AML, particularly through combinations including venetoclax and menin inhibitors, MRD-directed treatment, and post-remission maintenance approaches. As summarized in Table 2, a phase III trial (NCT06852222) is evaluating venetoclax plus azacitidine combined with the menin inhibitor bleximenib in newly diagnosed, unfit patients with NPM1 mutation or KMT2A rearrangement, with the aim of increasing MRD clearance and reducing the need for HSCT [34].

Similarly, NCT04867928 is assessing venetoclax plus azacitidine as MRD-guided therapy in patients with molecular relapse, where treatment response may influence the timing or necessity of allogeneic transplantation [34].

Early-phase studies combining venetoclax with menin inhibitors such as revumenib (NCT03013998) have demonstrated promising rates of molecular remission with manageable toxicity profiles [32,35].

In addition, the GIMEMA AML2521 trial is investigating venetoclax-based MRD-directed therapy to prevent or delay hematologic relapse in patients with molecular persistence after initial treatment [33].

Finally, the phase III study NCT07007312 is evaluating the addition of ziftomenib to standard regimens in untreated NPM1-mutated AML [36].

Collectively, these trials aim to deepen molecular responses, optimize the role of allogeneic HSCT, and define conditions under which treatment discontinuation (TFR) may be safely considered.

Other studies, such as NCT05184842 [37] or NCT04128501 [38], are currently in recruitment phases or not yet published, and the available information is limited. These trials are respectively exploring venetoclax combined with menin inhibitors, post-remission maintenance strategies, and novel immunotherapeutic or target-specific combinations in patients with refractory disease or molecular relapse. However, the specific inclusion of NPM1-mutated patients and preliminary data remain insufficiently documented to draw definitive conclusions.

In conclusion, the ongoing research pipeline holds promise to significantly expand therapeutic options for patients with NPM1-mutated AML. Future strategies are expected to move increasingly toward personalized therapy, adjusting intensity and duration of treatment based on MRD clearance and individual molecular characteristics. In this setting, treatment-free remission may become a realistic goal for an increasing number of patients.

## 7. Conclusions

Despite significant progress in the use of venetoclax in combination with HMAs for the management of NPM1-mutated AML, several critical issues remain that deserve attention. First of all, most of the available studies, although promising, are based on relatively small cohorts or retrospective data, which limits the generalizability of the results. The lack of long-term data makes it difficult to establish with certainty the durability of complete molecular remission and the real possibility of discontinuing treatment without compromising prognosis.

From a clinical standpoint, the management of side effects represents a significant challenge. Severe neutropenia and related infections are frequently reported, requiring careful monitoring and sometimes dose adjustments or temporary suspension of venetoclax. These aspects affect patients’ quality of life and treatment continuity, fundamental factors to consider, especially in elderly or frail patients.

Another unresolved issue concerns the optimal strategy for treatment duration and therapy discontinuation. Although recent studies suggest that discontinuation in patients with stable complete molecular response is possible, there is no unanimous consensus on precise criteria or ideal timing, and the risk of relapse remains a concern. This uncertainty reflects the need for prospective controlled studies dedicated to defining MRD-guided treatment algorithms.

Finally, debates persist on the role of allogeneic transplantation in this population. While transplantation remains the only potentially curative option, especially in the presence of high-risk co-mutations or persistent MRD, the efficacy of venetoclax in converting MRD status from positive to negative could reduce the need for aggressive procedures, at least in a selected subgroup of patients. This therapeutic evolution opens new scenarios but requires further evidence before it can contribute to clinical decision-making.

## Figures and Tables

**Figure 1 cancers-17-03733-f001:**
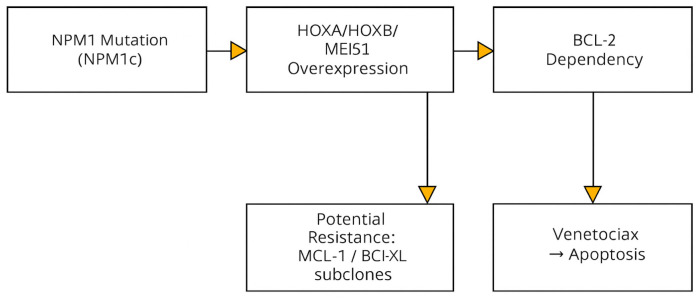
Molecular mechanisms of NPM1 mutation and BCL-2 inhibition.

**Figure 2 cancers-17-03733-f002:**
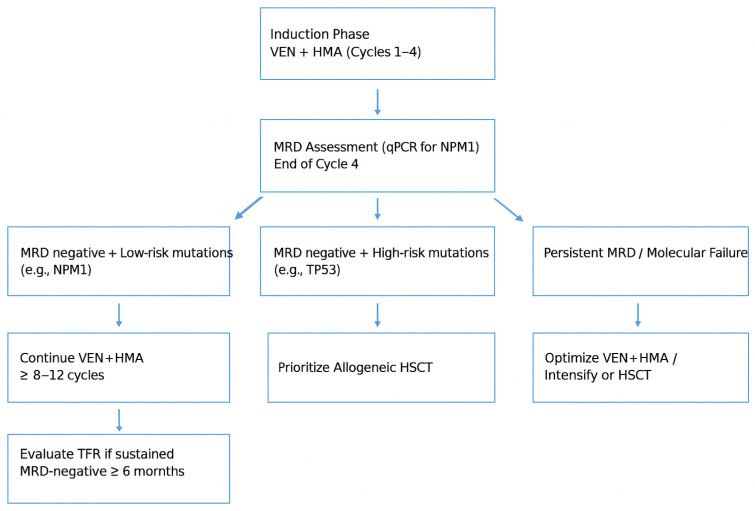
Illustrates the clinical continuum from induction therapy to MRD-guided management and treatment-free remission (TFR), emphasizing how molecular monitoring directs treatment decisions.

**Table 1 cancers-17-03733-t001:** Post-Remission Strategies in NPM1-Mutated Acute Myeloid Leukemia.

Post-Remission Strategy	Patient Indications	Advantages	Limitations/Risks	Supporting Evidence
Active Observation	MRD-negative patients without additional genetic risks	Avoids treatment toxicity; maintains good quality of life	Risk of relapse if MRD monitoring is inadequate	Retrospective study by Chua et al. [7], prospective study by DiNardo et al. [14]
Maintenance with Venetoclax	Patients in molecular remission without persistent MRD	Potential prolongation of remission duration	Lack of randomized trial evidence; empirical approach	Retrospective data from SEER, CIBMTR registries [10]; NCCN guidelines [20]
Allogeneic Transplant	MRD-positive patients or those with high-risk mutations	Potentially definitive treatment via graft-versus-leukemia effect	Transplant-related toxicity and mortality	NILG and Fraccaroli et al. studies [21]; EBMT guidelines [22]

**Table 2 cancers-17-03733-t002:** Ongoing Venetoclax-Based Trials in NPM1-AML.

Trial ID	Design/Phase	Population	Therapeutic Strategy	Status
NCT06852222	Phase III, randomized	NPM1-mutated or KMT2A-rearranged AML, untreated, unfit	Venetoclax + Azacitidine ± Menin inhibitor (Bleximenib)	Ongoing
NCT04867928	Phase II	NPM1-mutated AML in molecular relapse	Venetoclax + Azacitidine as bridge to transplant	Ongoing
NCT03013998	Phase II	NPM1/KMT2A AML, elderly patients	Venetoclax + Azacitidine + Menin inhibitor (Revumenib)	Ongoing
GIMEMA AML2521	Phase III	NPM1-mutated AML in molecular failure post-induction	Venetoclax + Azacitidine for MRD clearance and relapse prevention	Ongoing
NCT07007312	Phase III, randomized, placebo-controlled	NPM1-mutated or KMT2A-rearranged AML, newly diagnosed	Ziftomenib + Venetoclax + Azacitidine or 7 + 3	Ongoing
NCT05184842	Phase I	AML (including NPM1-mutated)	Venetoclax + Azacitidine + Menin inhibitor (SNDX-5613)	Recruiting (limited data)
NCT04128501	Phase II	AML (including NPM1 mutated), post-remission setting	Venetoclax maintenance post-induction	Recruiting

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
