# Peer review of "Harnessing Venetoclax in NPM1-Mutated AML: A Path to Sustained Remission and Beyond"

_cancers, 2025, doi:10.3390/cancers17233733_

Round 1

Reviewer 1 Report

Comments and Suggestions for Authors

I have reviewed the manuscript. This is a very nice overview on the venetoclax containing treatment options in NPM1-mutated AML. I have a few comments/suggestions.

1) Do the authors focus on isolated NPM1-mutated AML and only data regarding this patient population in this review? As the authors stated, some other mutations including FLT3-ITD may co-occur. I guess this should be clearly stated.  

2) When the term "MRD" is used, do the authors mean molecular MRD, flow MRD, or both? This should be again clearly stated and discussed if necessary.

Author Response

1)    Do the authors focus on isolated NPM1-mutated AML and only data regarding this patient population in this review? As the authors stated, some other mutations including FLT3-ITD may co-occur. I guess this should be clearly stated.

We thank the Reviewer for this important comment. We have now clarified that the focus of this review is primarily on isolated NPM1-mutated AML, where the benefit of venetoclax-based therapy is most clearly defined. We also now explicitly state that co-mutations such as FLT3-ITD, TP53, ASXL1, and RUNX1 may alter response patterns, and outcomes for these subgroups are therefore discussed separately (Introduction 54-63, Section 3 121-126 and 222-226).

2)    When the term "MRD" is used, do the authors mean molecular MRD, flow MRD, or both? This should be again clearly stated and discussed if necessary.

We thank the Reviewer for this important clarification. We have now specified in the Introduction that the term “MRD” refers to molecular MRD assessed by quantitative RT-PCR for NPM1, unless otherwise indicated. Mentions of flow cytometry–based MRD are explicitly noted where applicable (Introduction 78-80) 

Reviewer 2 Report

Comments and Suggestions for Authors

Molica et al. prepared a detailed review of available data with respect to treating NPM1-mutated AML in adults with Venetoclax.

Major observations for improvements:

1) Along with chapter 2. Molecular Rationale... a brief discussion about what is known regarding entire clone or even subclonal occurrence of NPM1 mutations in AML and potential relevance for Venetoclax efficacy is warranted. Figure 1 should be rebuilt with more academic content - less simplistic.

2) In chapter 3.1. between lines 128 - 150 there is a flaw of description of literature content: lines 128 - 132 describe content of ref [12] Othman et al. and not as cited of Chua et al. [7]. This must be corrected. In addition, lines 140 - 150 are redundantly describing similar content (see lines 131 - 132 versus 146 - 147), however, here with correct reference [12] Othman et al.

3) In chapter 3.2. lines 169-170: again, the reference to Chua et al. [7] is wrong but should be Othman et al. [12] - and the sentence is redundant with lines 144 - 145. Similar problem at lines 178 - 180: wrong reference to Chua et al. [7] while content is from Othman et al. [12] - and the sentence is redundant with lines 148 -150.

4) Figure 2 is too simplistic and does not encompass all potential situations or timeframes (dare a suggestion), like what are the choices to be made with respect to MRD response and when (after 4 cycles?), e.g. SZT or intensive chemo depending on age/fitness etc etc; how long to continue in minimum according to current knowledge...

5) In chapter 6, the bullet-point listing of trials (lines 314 - 338) is largely redundant with Table 2. This redundancy should be diminished.

6) in chapter 7, conclusions, there is statement on side effects (line 361), however, there is hardly any overview in the review on side effects of VEN treatments including HMA combinations. This gap must be closed and a dedicated chapter on side effects introduced.

Minor points:

1) Keywords are missing.

2) line 52, missing word "in": ... co-mutations such as in FMS-like tyrosine kinase 3 (FLT3).

3) line 120, missing words "in patients": ... benefit is the frontline treatment in patients who are eligible...

4) line 171, reference name is wrong: should be Jimenez-Chillon and not Dillon.

5) line 203, missing word "ratio of": ... in the absence of high-allelic ratio of FLT3-ITD...

6) line 261, missing word "ratio of": ... (e.g., high allelic ratio of FLT3-ITD, TP53,...

7) line 262, there are references missing to this statement.

8) line 306-307: this sentence is not well understandable: do you mean that AML with NPM1m is a chronic disease??? A rewording is necessary.

Author Response

1) Along with chapter 2. Molecular Rationale... a brief discussion about what is known regarding entire clone or even subclonal occurrence of NPM1 mutations in AML and potential relevance for Venetoclax efficacy is warranted. Figure 1 should be rebuilt with more academic content - less simplistic.

We thank the reviewer for this thoughtful comment. We have now added a concise clarification in Section 2 regarding the clonal versus subclonal occurrence of NPM1 mutations and its relevance for venetoclax sensitivity (133-137). This addition strengthens the biological rationale without altering the structure of the section. b)    We thank the reviewer for this helpful suggestion. Figure 1 has been revised to incorporate clonal architecture and mechanisms of resistance, providing a more comprehensive biological rationale for venetoclax sensitivity.

2)In chapter 3.1. between lines 128 - 150 there is a flaw of description of literature content: lines 128 - 132 describe content of ref [12] Othman et al. and not as cited of Chua et al. [7]. This must be corrected. In addition, lines 140 - 150 are redundantly describing similar content (see lines 131 - 132 versus 146 - 147), however, here with correct reference [12] Othman et al. We thank the reviewer for carefully identifying these inaccuracies and redundancies. We have now corrected the reference attribution and removed the repeated content to improve clarity and accuracy.

a)   Corrected Reference Attribution The results regarding MRD negativity rates, survival outcomes, and discontinuation feasibility in the multicenter cohort study have now been correctly attributed to Othman et al. [12], rather than to Chua et al. [7] (Lines 159-166)

b)    Elimination of Redundant Paragraph The paragraph beginning with “A particularly impactful recent study provides strong real-world evidence…” has been removed, as it repeated the same results from Othman et al. [12] already described earlier in the section (lines 173-183)

3)In chapter 3.2. lines 169-170: again, the reference to Chua et al. [7] is wrong but should be Othman et al. [12] - and the sentence is redundant with lines 144 - 145. Similar problem at lines 178 - 180: wrong reference to Chua et al. [7] while content is from Othman et al. [12] - and the sentence is redundant with lines 148 -150.

We thank the reviewer for carefully identifying these inaccuracies and redundancies. We have now corrected the reference attribution and removed the repeated content to improve clarity and accuracy. The duplicated summary of Othman et al.’s findings has been removed, and the citations referring incorrectly to Chua et al. in lines 201–203 and 213–215 have been corrected to Othman et al. [12].

4) Figure 2 is too simplistic and does not encompass all potential situations or timeframes (dare a suggestion), like what are the choices to be made with respect to MRD response and when (after 4 cycles?), e.g. SZT or intensive chemo depending on age/fitness etc etc; how long to continue in minimum according to current knowledge...

We thank the reviewer for this constructive suggestion. Figure 2 has been completely redesigned to provide a clearer and more comprehensive MRD-based decision algorithm. Specifically, we now: 1.    Distinguish between MRD-negative cases with low-risk molecular background (e.g., isolated NPM1) and MRD-negative cases with high-risk co-mutations (e.g., TP53). 2.    Explicitly indicate prioritization of allogeneic HSCT in MRD-negative patients harboring high-risk mutations. 3.    Clarify management strategies for persistent MRD or molecular failure, including treatment intensification or HSCT. 4.    Ensure graphical clarity by separating all boxes and placing arrows entirely outside the shapes, as requested. We believe the revised figure now provides a more academically informative and clinically applicable therapeutic pathway.

5) In chapter 6, the bullet-point listing of trials (lines 314 - 338) is largely redundant with Table 2. This redundancy should be diminished.

We have revised Section 6 to eliminate redundancy with Table 2. The previous bullet-point list has been replaced with a concise narrative synthesis, and all relevant clinical trials remain clearly cited. Table 2 now provides detailed trial characteristics, while the text highlights their conceptual significance. (line 440-461)

6) in chapter 7, conclusions, there is statement on side effects (line 361), however, there is hardly any overview in the review on side effects of VEN treatments including HMA combinations. This gap must be closed and a dedicated chapter on side effects introduced. We thank the reviewer for this important observation.

In the revised manuscript, we have added a dedicated section addressing the toxicity profile of venetoclax-based regimens. This new section (now Section 3.3. Toxicity and Adverse Events Associated with Venetoclax-Based Therapy) provides a concise but comprehensive overview of the most relevant hematologic and infectious complications, dose-modification strategies, and supportive care considerations. Additionally, the conclusion has been revised to include a short statement reinforcing the clinical relevance of treatment-associated toxicity.

Minor points: We thank the reviewer for this thoughtful comment. As requested, we have revised the sentence in the Conclusions section (formerly lines 383–386) to avoid the misleading implication that NPM1-mutated AML may represent a chronic disease. The sentence has now been reformulated to accurately reflect the concept of time-limited therapy in selected patients, without altering the acute and relapsing nature of the disease. In addition, we have implemented all minor revisions requested.

Reviewer 3 Report

Comments and Suggestions for Authors

The manuscript presents a comprehensive and well-structured review of the role of venetoclax in acute myeloid leukemia (AML) with NPM1 mutation. The text is clearly written, up to date with relevant references (through 2025), and effectively integrates the biological, clinical, and therapeutic aspects of the topic. The focus on minimal residual disease (MRD) as a tool for therapeutic personalization and the discussion of treatment discontinuation strategies (treatment-free remission, TFR) are particularly commendable.

However, the work would benefit from greater analytical and critical depth, especially in the discussion of resistance mechanisms, methodological limitations of the studies reviewed, and the balance between efficacy and toxicity of venetoclax-based regimens. In addition, the abstract and introduction repeat some ideas and could be more concise.

  • A key limitation identified in the manuscript is that the encouraging clinical outcomes with venetoclax-based regimens in NPM1-mutated AML are mainly supported by studies with significant methodological constraints. The authors acknowledge that most available evidence arises from retrospective analyses or small patient cohorts, which restricts the generalizability of the conclusions and highlight the need for randomized, large-scale trials. Moreover, the lack of long-term follow-up data prevents a clear understanding of remission durability and the long-term safety of treatment discontinuation.

Given these methodological limitations, do the authors foresee any ongoing or planned prospective randomized studies that could validate the durability and safety of venetoclax discontinuation in NPM1-mutated AML?

  • How do the authors suggest optimizing the management of venetoclax-related toxicities, such as neutropenia and infections, to maintain therapeutic efficacy while minimizing interruptions, especially in vulnerable patient populations?
  • The review presents an optimistic narrative about an emerging “paradigm shift” toward less toxic, targeted therapies and the potential achievement of treatment-free remission (TFR) in NPM1-mutated AML. However, it simultaneously and repeatedly concedes that this paradigm is largely supported by limited, primarily retrospective evidence. Fundamental questions regarding optimal treatment duration, maintenance strategies, and safe discontinuation criteria remain unresolved. This contrast underscores a critical gap between the envisioned therapeutic future and the current evidence base, which still lacks robust prospective data to establish standardized clinical algorithms.

    Given this gap between expectation and evidence, how do the authors propose to balance enthusiasm for TFR and targeted therapy approaches with the need for caution until high-quality, prospective data become available?

  • Considering the current lack of prospective evidence, what specific clinical or molecular criteria do the authors believe could help identify patients who might truly benefit from venetoclax maintenance therapy, rather than simple observation?

Author Response

We thank the Reviewer for this insightful comment. We have expanded Section 5 by adding a detailed discussion of the ongoing prospective studies aimed at evaluating the safety and patient selection criteria for venetoclax discontinuation (NCT04867928, GIMEMA AML2521, NCT06852222). (Lines 388-397) In addition, we have revised Section 3.3 to provide a clearer description of clinical strategies for managing venetoclax-related neutropenia and infections, with the goal of optimizing treatment continuity, particularly in more vulnerable patient populations (lines 245-252)

We have added a paragraph at the end of Section 5 discussing the need to balance enthusiasm for TFR approaches with appropriate clinical caution based on current evidence (lines 399-407), and expanded Section 4.2 to specify clinical and molecular criteria that may guide the selection of patients for venetoclax maintenance versus observation. These revisions clarify our position and directly address the considerations raised (Lines 304-313). 

Round 2

Reviewer 2 Report

Comments and Suggestions for Authors

Molica et al. have nicely revised their first version - a few modifications, however, should still be done:

1) The sentence "In clinical practice, ... with recurrent infections." (lines 248-252) is largely redundant with line 235 - 239. I suggest to delete the quoted sentence line 248-252 or include parts in the first sentence lines 248-252 as felt necessary.

2) New sentence line 387-390 (version with track changes shown) "Although controlled prospective studies...." I suggest a modification: "....venetoclax-based treatment may be so highly effective that it can also be planned time-limited, thereby modifying the therapeutic paradigm that any treatment interruption is associated with a high risk of relapse."

3) New paragraphs in chapter 5: lines 392-394 are redundant with lines 403-406. I suggest to delete the sentences "While the emerging evidence..." until "...remains incompletely defined." (i.e. in lines 403-406)

Author Response

  1. done
  2. done
  3. done

Reviewer 3 Report

Comments and Suggestions for Authors

The revised version adequately addresses the reviewers’ comments and suggestions, resulting in a clearer, more balanced, and analytically sound discussion. The manuscript now provides a well-structured and insightful overview of the topic, with improved critical depth regarding the methodological limitations, resistance mechanisms, and therapeutic perspectives

Author Response

thank you